# On the Statistical Discrepancy and Affinity of Priority Vector Heuristics in Pairwise-Comparison-Based Methods

**DOI:** 10.3390/e23091150

**Published:** 2021-09-01

**Authors:** Pawel Tadeusz Kazibudzki

**Affiliations:** Faculty of Economics and Management, Opole University of Technology, St. Luboszycka 7, 45-036 Opole, Poland; p.kazibudzki@po.edu.pl

**Keywords:** pairwise comparisons, ranking, prioritization, priorities deriving methods, AHP, Monte Carlo simulations

## Abstract

There are numerous priority deriving methods (PDMs) for pairwise-comparison-based (PCB) problems. They are often examined within the Analytic Hierarchy Process (AHP), which applies the Principal Right Eigenvalue Method (PREV) in the process of prioritizing alternatives. It is known that when decision makers (DMs) are consistent with their preferences when making evaluations concerning various decision options, all available PDMs result in the same priority vector (PV). However, when the evaluations of DMs are inconsistent and their preferences concerning alternative solutions to a particular problem are not transitive (cardinally), the outcomes are often different. This research study examines selected PDMs in relation to their ranking credibility, which is assessed by relevant statistical measures. These measures determine the approximation quality of the selected PDMs. The examined estimates refer to the inconsistency of various Pairwise Comparison Matrices (PCMs)—i.e., ***W*** = (*w_ij_*),* w_ij_* > 0, where *i*, *j* = 1,…, *n*—which are obtained during the pairwise comparison simulation process examined with the application of Wolfram’s Mathematica Software. Thus, theoretical considerations are accompanied by Monte Carlo simulations that apply various scenarios for the PCM perturbation process and are designed for hypothetical three-level AHP frameworks. The examination results show the similarities and discrepancies among the examined PDMs from the perspective of their quality, which enriches the state of knowledge about the examined PCB prioritization methodology and provides further prospective opportunities.

## 1. Introduction

The method of creating a ranking based on pairwise comparisons of alternatives was already known in the Middle Ages. The first work on this subject was probably that by Ramon Lull [1], who described election processes based on comparisons of mutual alternatives. Over time, other research on the pairwise comparisons method appeared; e.g., studies on electoral systems, such as the Condorcet method and the Copeland method, and many others on social choice and welfare systems [2]. In time, alternatives began to be compared quantitatively, which was initially connected with the need to compare psychophysical stimuli [3,4]. This path was later developed [5] and used in various forms for different objectives, including economics [6], consumer research, psychometrics, health care and others. Thanks to Saaty and his seminal paper [7] in which he defined the Analytic Hierarchy Process (AHP), comparing alternatives in a pairwise mode began to be considered basically as a multi-criteria decision-making method. The undisputable success of the AHP is probably due to the fact that Saaty proposed a complete solution including a ranking calculation algorithm, an inconsistency index as a method of determining data quality and a hierarchical model allowing decision makers (DMs) to handle multiple criteria [8,9,10,11]. Overtime, numerous studies have presented scientific evidence of the fundamental flows of the AHP; see, e.g., [12,13,14,15].

However, different research studies concerning the pairwise comparison method have resulted in many priority deriving methods (PDMs) which, for brevity, are not discussed in this article in detail; however, the interested reader may want to find references in which these methods are scrutinized (e.g., [16,17,18,19,20,21,22,23,24,25,26,27,28,29,30,31,32,33,34,35,36,37,38,39,40,41,42,43,44,45,46,47,48]). In addition, various inconsistency approaches for human judgments were devised, which also are not discussed in this paper in detail as they are not within the scope of this research. It seems probable that the two most popular PDMs for PCB ranking problems are the Principal Right Eigenvalue Method (PREV), proposed by Saaty [12,49,50] and the Logarithmic Least Squared Method (LLSM), also known as Geometric Mean Method (GMM), devised by Crawford and Williams [21,22]. A further well-known PDM is the Simple Normalized Column Sum Procedure (SNCS), proposed by Zahedi [47] and promoted, e.g., by Choo and Wedley [19] and Saaty [51]. Indeed, an underestimated and overlooked method in the literature is the last PDM selected for this research due to its features; i.e., the Logarithmic Squared Deviations Minimization Method (LSDM), elaborated and evaluated by Kazibudzki [30,52].

It is easy to verify that in the case of consistent human judgments that provide cardinally transitive Pairwise Comparison Matrices (PCM), all PDMs lead to the same solution. However, when inconsistent PCMs must be taken into account, the resulting rankings differ from each other. This research is part of the discussion of the properties of various PDMs applied for pairwise-comparison-based (PCB) problems, which are often examined with the application of the AHP [53]. Despite the large number of publications on the topic, this issue is considered inspiring and challenging as it seems that pairwise comparisons could lead to credible measures of DM preferences. Thus, deriving true priority vectors (PVs) from intuitive pairwise comparisons of decision makers (DMs) is also a crucial issue for the multiple criteria decision-making (MCDM) concept based on the AHP. Noticeably, the standard AHP applications commonly utilize PREV because it was derived mathematically by the creator of the AHP, who considered it the only correct solution for PCB problems [42,54,55,56]. 

The objective of this scientific research is to examine the similarities and discrepancies of a few selected PDMs in order to determine their suitability for PCB problems in relation to their ranking credibility. Thus, it was decided to apply Monte Carlo simulations (MCS) for this purpose. However, rather than simulating and analyzing results for a single PCM, as has been done thus far by many other authors, it was decided to design simulation scenarios and analyze their outcomes from the simple multi-criteria decision perspective, examined via the most common AHP framework, which thus far has been undertaken by only a few authors; e.g., [30,57]. As such, it is assumed that the three-level AHP model—alternatives, criteria and goal—is considered, which is assumed to deal with the hypothetical decision problems. Then, the simulation results for selected PDMs are compared. The examination results present the effect of various scenarios within the simulation process and reflect human judgment errors during pairwise comparisons. Thus, selected PDMs are examined from the perspective of their ranking credibility, which is evaluated with the application of a few available statistical measures; e.g., the Mean Spearman Rank Correlation Coefficient (MSRC), Mean Pearson Correlation Coefficient (MPCC) and Mean Average Absolute Deviation (MAAD). These measures determine the differences in the quality of PV estimation, understood as a true ranking preservation capability of the selected PDM, and are selected intentionally while considering that other non-statistical compatibility indices exist in the literature: e.g., the Garuti index [58] or Saaty compatibility index [59]. Certainly, different MCDM methods may yield different results when applied to the same problem [60], and that is why a single method application, such as AHP, can also lead to different priorities. This phenomenon has been already studied previously, mainly by focusing on the ranks that different priorities imply [61]. This research paper aims to extend the focus of this study from the ordinal to a cardinal focus, where differences in the ranks are considered. That is why emphasis is placed on the rank preservation phenomenon instead of the closeness of obtained priority vectors (PVs) as it has applicability during compatibility analysis.

Given the reality of our physical world, no study is perfect. In order to compare the characteristics of the estimates obtained in the simulation process for the selected PDM, various scenarios were simulated in relation to different sources of PCM inconsistency. Fundamentally, PCM inconsistency commonly results from errors caused by the nature of human judgments and errors due to the technical realization of the pairwise comparison procedure; i.e., rounding errors and errors resulting from the forced reciprocity requirement commonly imposed in PCB ranking problems. All the above errors can be simulated, but the nature of human judgments can be represented only as the realization of some random process in accordance with the assumed probability distribution of the perturbation factor; e.g., uniform, gamma, truncated normal and log-normal. As this is only a stochastic process generated by a computer, it constitutes a certain limitation of the presented research.

The research paper is structured as follows: firstly, preliminaries about pairwisecomparison-based problems are presented (Section 2); then, the examination methodology is introduced and exemplified (Section 3) in two subsections: Section 3.1 is devoted to the concept design with preliminary results, and Section 3.2 introduces the target examination scenario and presents further examples of examination results. The results of the complete examination and a discussion are presented in Section 4, leading to our research conclusions (Section 5); finally, we end the paper with final remarks.

## 2. Preliminaries about PCB Ranking Problems

The conventional PDM in the AHP is founded on the mathematical structure of consistent PCMs and the related capability of PREV to produce actual or approximate weights. 

Oscar Perron proved that if ***W*** = (*w_ij_*),* w_ij_* > 0, where *i*, *j* = 1,…, *n*, then ***W*** has a simple positive eigenvalue λ_max_ called the principal eigenvalue of ***W***, and λ_max_|λ_k_| for the remaining eigenvalues of ***W***. Moreover, the principal right eigenvector ***w*** = [*w*_1_,…, *w_n_*]^T^, which is a solution of ***Ww*** = λ_max_***w*,** has *w_i_* > 0, *i* = 1,…, *n*. If the relative weights of a set of activities are known, they can be expressed as PCMs. 

If we know *W*(*w*) but do not know *w*, we can use Perron’s theorem to solve this problem for *w*. The solution leads to *n* unique values for *lambda*, with a bounded vector *w* for each of the *n* values. The PCM matrix in the AHP reflects the relative weights of the actions considered (criteria, scenarios, players, alternatives, etc.), so the matrix *W*(*w*) has a particular shape. Each subsequent row of this matrix is a constant multiple of the first row. In this case, the matrix *W*(*w*) has only one non-zero eigenvalue, and since the sum of the eigenvalues of a positive matrix is equal to the sum of its diagonal elements, the single non-zero eigenvalue in this case is equal to the size of the matrix and can be denoted as *lambda*(*max*) = *n*. The norm of the vector *w* can be written as $\|w\|=e^{T}w$, where *e* = [1, 1,..., 1]^T^ and *w* can be normalized by dividing by its norm. For the sake of clarity, *w* is further specified in its normalized form.

Taking the above into consideration, the conventional concept of a PDM in the AHP can be presented as follows:(1)$$\left[\begin{array}{ccccc} w_{1}/w_{1} & w_{1}/w_{2} & w_{1}/w_{3} & \ldots & w_{1}/w_{n}\\ w_{2}/w_{1} & w_{2}/w_{2} & w_{2}/w_{3} & \ldots & w_{2}/w_{n}\\ w_{3}/w_{1} & w_{3}/w_{2} & w_{3}/w_{3} & \ldots & w_{3}/w_{n}\\ \vdots & \vdots & \vdots & & \vdots \\ w_{n}/w_{1} & w_{n}/w_{2} & w_{n}/w_{3} & \ldots & w_{n}/w_{n} \end{array}\right]\times \left[\begin{array}{c} w_{1}\\ w_{2}\\ w_{3}\\ \vdots \\ w_{n} \end{array}\right]=n\cdot \left[\begin{array}{c} w_{1}\\ w_{2}\\ w_{3}\\ \vdots \\ w_{n} \end{array}\right]$$

Thus, the following definitions (D) can be also presented:**D[1]**: If the matrix ***W***(*w*) elements satisfy the prerequisite *w_ij_* = 1/*w_ji_* for all *i*, *j* = 1,…, *n*, then the matrix ***W***(*w*) is called *reciprocal*.**D[2]**: If the following assumptions are true: (a) if for any *i* = 1,…, *n*, an element *w_ij_* is not less than an element *w_ik_*, then $w_{ij}\geq w_{ik}$ for *i* = 1,…, *n*, and (b) if for any *i* = 1,…, *n*, an element *w_ji_* is not less than an element *w_ki_*, then $w_{ji}\geq w_{ki}$ for *i* = 1,…, *n*, and the matrix ***W***(*w*) is called *ordinal transitive*,**D[3]**: If the elements of a matrix ***W***(*w*) satisfy the condition *w_ik_w_kj_* = *w_ij_* for all *i*, *j*, *k* = 1,…, *n*, and the matrix is *reciprocal*, then it is called *consistent* or *cardinal transitive*.

Certainly, when the AHP is utilized, a ***W***(*w*) which would reflect the true weights given by the actual PV is unknown. 

Since the human mind is not an accurate measuring device, it does not give accurate results in tasks such as the following: “compare—using a given ratio scale—your preferences for alternative 1 and alternative 2”. Thus, *W*(*w*) is not known, but only its estimate *X*(*x*) containing intuitive judgments that are more or less close to *W*(*w*) depending on the DM’s individual taste, specific knowledge, experience, ability and even momentary mood or frame of mind. In such cases, the consistency property does not apply, and the conventional notion of the PDM in the AHP is no longer applicable. However, it has been shown that for any PCM, small perturbations in the items entail similar perturbations in the eigenvalues, so Perron’s theorem can be used to estimate the true PV. Then, instead of matrix equation (1), the solution of matrix equation (2) gives us *w* as the right principal eigenvector (PREV) associated with λ_max_.
(2)$$\left[\begin{array}{ccccc} x_{1}/x_{1} & x_{1}/x_{2} & x_{1}/x_{3} & \ldots & x_{1}/x_{n}\\ x_{2}/x_{1} & x_{2}/x_{2} & x_{2}/x_{3} & \ldots & x_{2}/x_{n}\\ x_{3}/x_{1} & x_{3}/x_{2} & x_{3}/x_{3} & \ldots & x_{3}/x_{n}\\ \vdots & \vdots & \vdots & & \vdots \\ x_{n}/x_{1} & x_{n}/x_{2} & x_{n}/x_{3} & \ldots & x_{n}/x_{n} \end{array}\right]\times \left[\begin{array}{c} w_{1}\\ w_{2}\\ w_{3}\\ \vdots \\ w_{n} \end{array}\right]=\lambda_{\max}\left[\begin{array}{c} w_{1}\\ w_{2}\\ w_{3}\\ \vdots \\ w_{n} \end{array}\right]$$

In practice, the solution of PREV is obtained by raising the matrix *X*(*x*) to a sufficiently large power, then summing the rows of *X*(*x*) and normalizing the resulting vector to obtain a vector *w*, which can be expressed by the following formula:(3)$$w=\underset{k\to \infty }{\lim}\left(\frac{X^{k}(x)\times e}{e^{T}\times X^{k}(x)\times e}\right)$$
where *e* = [1, 1,…, 1]^T^.

Noticeably, the relation between elements of ***X***(*x*) and ***W***(*w*) can be expressed in the form of the following formula:(4)$$x_{ij}=e_{ij}w_{ij}$$
where *e_ij_* is a disturbance coefficient oscillating close to unity; e.g., *e_ij_*∈[0.5, 1.5]. It is important to emphasize that in the statistical approach, *e_ij_* reflects the realization of a random variable with a provided probability distribution (PD) that can be modeled and that reflects imperfect human pairwise comparisons. In the literature, the following types of PDs are often considered for different implementation purposes: *gamma, log-normal, truncated-normal or uniform* [36,47]. However, in addition to these most popular types of PDs, one can also find applications of Cauchy, Laplace, *triangular* or *beta* PDs [24] and Fisher–Snedecor PDs, which were recently introduced by Kazibudzki [30]. Usually, the maximal spread for $e_{ij}\in \left[0.01,\;1.99\right]$, which may be perceived as strange as this interval is highly asymmetric. However, it is perfectly reasonable to have asymmetric intervals for perturbation factors as DM judgments are reflected with the application of a particular numerical scale whose numbers are usually not higher than nine (Saaty’s scale). So, multiplying five by more than two, for example, is simply pointless, as the result must still be rounded to the nearest value of the scale; e.g., Saaty’s scale, whose maximal value equals nine. On the other hand, multiplying 9 by 0.01 gives 0.09, which can be naturally rounded to 0.1(1) and may reflect a reversal phenomenon of preferences, for example. However, the symmetric interval for a perturbation factor with a seminal proposed PD and its application outcome is also presented in this research in Section 4.

In general, the discrepancy of the perturbed PCM reflects the results of errors caused by the nature of human judgment and errors due to the technical implementation of the pairwise matching procedure. The latter are mainly due to rounding errors and errors resulting from forced reciprocity requirements. Rounding errors are related to the numerical ratio scale, the values of which must be used by future DMs to express their judgments in a certain way [25,62,63,64]. In common AHP applications, the Saaty numerical scale, consisting of integers from 1 to 9 and their inverse values, is by far the most popular. However, other scales are also known [25,65,66,67,68,69,70], such as the geometric scale, for which the linguistic variables of the Saaty scale have different numerical values; the most common, and thus the approach used in this study, is 2^n/2^, where *n* includes integers from minus 8 to 8, but an arbitrarily defined numerical scale including integers from 1 to n and their inverse values is also possible.

The basic concept of the AHP certainly attracts attention and as such is being developed; see, e.g., [37,71,72,73,74,75,76]. At the same time, it is argued that as long as inconsistency in pairwise comparisons is allowed, PREV is the fundamental theoretical concept for ranking PCB problems and no other PDM matches it. At the same time, a number of other PDMs have been proposed over the past three decades, starting with the most popular LLSM [21,22] and other methods based on optimization models with constraints (see [28,52,62,77] ), including the least-squares method [36,47] and various versions of goal programming methods (see [18,19,27,78,79,80]), as well as methods based on some statistical concepts (see [12,16,34,46,61,81,82]), methods based on fuzzy preference descriptions (see [48,72,83]) and heuristic algorithms (see [33,35,84,85]).

It has been argued that the primary AHP–PDM—i.e., the PREV method—is necessary and sufficient for unambiguous ranking with the ratio scale inherent in inconsistent pairwise comparisons [42,50,54,55,86,87]. However, this approach has also been heavily criticized; see, for example, [13,30,63,80,88,89,90,91,92]. Therefore, there are optional PDMs that differ from the basic PDM concept. Many of these are based on optimization and finding the vector *w* as a solution to the minimization problem given by the formula
min ***D***(***X***(*x*), ***W***(*w*))(5)
with some accompanying constraints such as positive coefficients and the normalization condition. Since the distance function ***D*** measures the interval between the matrices ***X****(x)* and ***W***(*w*), different ways of defining this lead to different PDM estimation results. Chu et al. in [19] describe and compare 18 PDMs, although some authors suggest that only 15 are distinctive. Undoubtedly, several other PDMs have appeared in the literature since Chu et al. [19] published their study; see, for example, [29,32,33,34,93,94,95]. Obviously, if the PCM is consistent, then all known PDMs match, although they do not guarantee that the resulting PV is error-free; see, e.g., [12,45,70,96]. However, in real situations, as noted earlier, human judgments inevitably lead to inconsistent PDMs, since inconsistency is a natural consequence of the dynamics of the human mind as well as a consequence of query methodology, incorrect inputs of judgment values and scaling procedures (i.e., rounding errors).

In this research study, apart from PREV, three optional PDMs are examined. They are defined by the formulae presented in Table 1.

## 3. Examination Methodology

### 3.1. Concept Design with Preliminary Results

The first step in PCB problems using the AHP is to create a hierarchy by breaking the specific problem into its major components. The basic AHP scenario includes a goal (an expression of the overall objective), criteria (factors to be considered in arriving at the final choice) and alternatives (feasible alternatives to achieve the final objective). Thus, the most basic AHP decision model consists of an objective–criteria–alternatives sequence. Therefore, this study adopts a basic three-level hierarchy including three criteria and three alternatives within each criterion.

The intent of this research is to examine the performance of PREV against the background of the performance of other selected PDMs available for PCB problems elaborated within the AHP. In order to achieve this objective, Monte Carlo simulations (MCS) were applied, but not as commonly performed; i.e., dedicated to a single PCM. This research involves an MCS scenario that encompasses the entire goal–criteria–alternatives sequence of the AHP, which is supposed to reflect the hypothetical PCB decisional problem (see the examples presented hereafter (Examples 1A and 1B)).

Firstly, the examination framework is presented in its simplified version as a methodological example. Thus, only technical distortions are used resulting from rounding errors during the application of Saaty’s scale and standard requirements of the AHP; i.e., forced reciprocity is demonstrated in the following hypothetical AHP model with three levels (a goal, four criteria and four alternatives). This model assumes that relative ratios of some physical attributes of certain objects are predetermined, and thus $H_{G}PV_{C}$, $H_{C2}^{C1}PV_{A}$ and $H_{C4}^{C3}PV_{A}$ are known. On the basis of the provided PV elements, the respective PCMs are reconstructed as shown in Equation (1). 


**Example 1A:**


$H_{G}PV_{C}$, and its related PCM denoting the weights quotients of $H_{G}PV_{C}$, reflecting the pairwise comparison results of criteria with respect to the goal:(6)$$\begin{array}{cc} & \begin{array}{cccc} C1\;\;\; & \;\;C2 & C3 & \;\;\;\;C4 \end{array}\\ \begin{array}{c} C1\\ C2\\ C3\\ C4 \end{array} & \left[\begin{array}{cccc} 1 & 1.4 & 3.5 & 1.16667\\ 0.714286 & 1 & 2.5 & 0.833333\\ 0.285714 & 0.4 & 1 & 0.333333\\ 0.857143 & 1.2 & 3 & 1 \end{array}\right] \end{array}\begin{array}{cc} & \begin{array}{c} H_{G}PV_{C}\\ \left[\begin{array}{c} 0.35\\ 0.25\\ 0.10\\ 0.30 \end{array}\right] \end{array} \end{array}$$$H_{C2}^{C1}PV_{A}$, and its related PCM denoting the weights quotients of $H_{C2}^{C1}PV_{A}$, reflecting the pairwise comparison results of alternatives with respect to criteria *C*1*–C*2:(7)$$\begin{array}{cc} & \begin{array}{cccc} \;\;\;\;\;\;A1\;\; & \;\;\;A2\;\;\; & \;A3\;\;\; & \;\;A4 \end{array}\\ \begin{array}{c} A1\\ A2\\ A3\\ A4 \end{array} & \left[\begin{array}{cccc} 1 & 1.4 & 2.33333 & 1.4\\ 0.714286 & 1 & 1.66667 & 1\\ 0.428571 & 0.6 & 1 & 0.6\\ 0.714286 & 1 & 1.66667 & 1 \end{array}\right] \end{array}\begin{array}{cc} & \begin{array}{c} H_{C2}^{C1}PV_{A}\\ \left[\begin{array}{c} 0.35\\ 0.25\\ 0.15\\ 0.25 \end{array}\right] \end{array} \end{array}$$$H_{C4}^{C3}PV_{A}$, and its related PCM denoting the weights quotients of $H_{C4}^{C3}PV_{A}$, reflecting the pairwise comparison results of alternatives with respect to criteria *C*3*–C*4:(8)$$\begin{array}{cc} & \begin{array}{cccc} A1\;\; & \;\;\;A2\;\;\;\;\;\; & \;\;\;A3\;\; & \;\;\;\;\;\;A4 \end{array}\\ \begin{array}{c} A1\\ A2\\ A3\\ A4 \end{array} & \left[\begin{array}{cccc} 1 & 0.666667 & 0.285714 & 0.25\\ 1.5 & 1 & 0.428571 & 0.375\\ 3.5 & 2.33333 & 1 & 0.875\\ 4 & 2.66667 & 1.14286 & 1 \end{array}\right] \end{array}\begin{array}{cc} & \begin{array}{c} H_{C4}^{C3}PV_{A}\\ \left[\begin{array}{c} 0.10\\ 0.15\\ 0.35\\ 0.40 \end{array}\right] \end{array} \end{array}$$
where $H_{G}PV_{A}$, $H_{C2}^{C1}PV_{A}$ and $H_{C4}^{C3}PV_{A}$ denote partial hypothetical PVs in the model.

After standard AHP synthesis, the hypothetical total PV (HTPV) is obtained and given as HTPV = [0.25, 0.21, 0.23, 0.31]^T^. Next, following the simplified version of the MCS scenario in this example, each PCM in the presented framework is perturbed. For simplicity of illustration, only two kinds of prospective PCM distortions are applied: rounding errors (each element of the particular PCM is rounded to Saaty’s scale) and reciprocity imposition errors (the PCM is transformed to be reciprocal in the way that only elements above its diagonal are taken into consideration, and elements below its diagonal are replaced by the inverses of their counterparts from above its diagonal). Next, on the basis of each PCM being perturbed in the above way, respective partial PVs (PPV_PREV_) are obtained with the application of the selected PDM—i.e., PREV. Finally, the total computed PV (TCPV_PREV_) for the exemplary model of the AHP is obtained (see Example 1B).


**Example 1B:**


PCM with criteria weights designated with respect to the goal and $PPV_{PREV}^{GOAL}$, computed on the basis of this PCM with the application of PREV:(9)$$\begin{array}{cc} & \begin{array}{cccc} C1 & C2 & C3 & C4 \end{array}\\ \begin{array}{c} C1\\ C2\\ C3\\ C4 \end{array} & \left[\begin{array}{cccc} 1 & 1 & 3 & 1\\ 1 & 1 & 2 & 1\\ 1/3 & 1/2 & 1 & 1/3\\ 1 & 1 & 3 & 1 \end{array}\right] \end{array}\begin{array}{cc} & \begin{array}{c} PPV_{PREV}^{GOAL}\\ \left[\begin{array}{c} 0.304999\\ 0.276859\\ 0.113143\\ 0.304999 \end{array}\right] \end{array} \end{array}$$
PCM with weights of alternatives designated with respect to criteria *C*1–*C*2 and $PPV_{PREV}^{C1C2}$, computed on the basis of this PCM with the application of PREV:(10)$$\begin{array}{cc} & \begin{array}{cccc} A1 & A2 & A3 & A4 \end{array}\\ \begin{array}{c} A1\\ A2\\ A3\\ A4 \end{array} & \left[\begin{array}{cccc} 1 & 1 & 2 & 1\\ 1 & 1 & 2 & 1\\ 1/2 & 1/2 & 1 & 1/2\\ 1 & 1 & 2 & 1 \end{array}\right] \end{array}\begin{array}{cc} & \begin{array}{c} PPV_{PREV}^{C1C2}\\ \left[\begin{array}{c} 0.285714\\ 0.285714\\ 0.142857\\ 0.285714 \end{array}\right] \end{array} \end{array}$$
PCM with weights of alternatives designated with respect to criteria *C*3–*C*4 and $PPV_{PREV}^{C3C4}$, computed on the basis of this PCM with the application of PREV:(11)$$\begin{array}{cc} & \begin{array}{cccc} A1 & A2 & A3 & A4 \end{array}\\ \begin{array}{c} A1\\ A2\\ A3\\ A4 \end{array} & \left[\begin{array}{cccc} 1 & 1/2 & 1/4 & 1/4\\ 2 & 1 & 1/2 & 1/3\\ 4 & 2 & 1 & 1\\ 4 & 3 & 1 & 1 \end{array}\right] \end{array}\begin{array}{cc} & \begin{array}{c} PPV_{PREV}^{C3C4}\\ \left[\begin{array}{c} 0.0887547\\ 0.1611320\\ 0.3550190\\ 0.3950950 \end{array}\right] \end{array} \end{array}$$

After standard AHP synthesis, the following result is obtained: TCPV_PREV_ = [0.2034, 0.2336, 0.2316, 0.3315]^T^, which is different from HTPV = [0.25, 0.21, 0.23, 0.31]^T^. The comparison of HTPV with its estimate TCPV_PREV_ enables selected statistical measures to be used—i.e., Spearman Ranks Correlation Coefficient (SRCC), Pearson Correlation Coefficient (PCC) and Mean Absolute Deviation (MAD)—which reflect the approximation quality of PREV. Mean values of the above measures were examined during the MCS in this research; the formulae for these are provided in Table 2. 

For the above illustrative values of HTPV and TCPV_PREV_, the presented measures are as follows: SRCC = 0.2, PCC = 0.8142, MAD = 0.023325. Noticeably, the comparison of the approximation quality of any PDM available for PCB ranking problems is possible in this way. Thus, it is also possible to examine the selected PDM for the AHP. The MCS designed for this purpose—i.e., processing the algorithm which exactly emulated the above explained steps 10,000 times—provided the scores shown in Table 3 and Table 4.

Considering the results presented in Table 3 and Table 4, it should be noticed that from the perspective of rank, the preservation capability of PREV is slightly lower than that of LSDM. It also can be noticed that LSDM performs slightly worse than PREV from the perspective of MPCC and MAAD values. The performance of other examined PDMs is slightly worse in both scenarios and from the perspective of all performance measures taken into consideration during this study.

### 3.2. Research Target Scenario Analysis with Further Results

Hereafter, the examination scenario is exemplified in its target version. Thus, not only are technical distortions resulting from rounding errors during the application of Saaty’s scale and standard requirements of the AHP will be used—i.e., forced reciprocity is demonstrated—but also human judgment errors are considered in the hypothetical AHP model with three levels (goal, four criteria and four alternatives) earlier depicted as in Example 1A. Hence, it is still assumed that relative ratios of some physical attributes of certain objects are predetermined; thus, $H_{G}PV_{C}$, $H_{C2}^{C1}PV_{A}$ and $H_{C4}^{C3}PV_{A}$ are known, and their respective PCMs are computed as in Equation (1). For the reader’s convenience, the model is duplicated herein and renamed as Example 2A for reference hereafter.


**Example 2A:**


For the assumed $H_{G}PV_{C}$ and its related PCMs representing the weights quotients of $H_{G}PV_{C}$, reflecting the pairwise comparison results of criteria with respect to the goal, see Equation (6). For the assumed $H_{C2}^{C1}PV_{A}$ and its related PCMs denoting the weights quotients of $H_{C2}^{C1}PV_{A}$, reflecting the pairwise comparison results of alternatives with respect to criteria *C*1–*C*2, see Equation (7). For the assumed $H_{C4}^{C3}PV_{A}$ and its related PCM denoting the weights quotients of $H_{C4}^{C3}PV_{A}$, reflecting the pairwise comparison results of alternatives with respect to criteria *C*3–*C*4, see Equation (8).

Next, following the target version of the MCS scenario in this example, each PCM in the presented framework of Example 2A is perturbed. This time, three kinds of prospective PCM distortions are applied: human judgment errors reflected by the applied perturbation factor *e_ij_* = 0.5, rounding errors (each element of the particular PCM is rounded to Saaty’s scale) and reciprocity imposition errors (the PCM is transformed to be reciprocal in a way that means only elements from above its diagonal are taken into consideration, and elements below its diagonal are replaced by inverses of their counterparts from above its diagonal). On the basis of each PCM being perturbed in the above way, respective partial PVs (PPV_PREV_) are obtained with application of the selected PDM; i.e., PREV. Finally, the total computed PV (TCPV_PREV_) for the exemplary model of the AHP is obtained; see Examples 2B–2D.


**Example 2B:**


Pairwise comparison results of criteria with respect to the goal after the implementation of the perturbation factor *e_ij_* = 0.5:(12)$$\begin{array}{cc} & \begin{array}{cccc} \;\;C1\;\; & \;C2\; & C3\;\; & \;\;C4 \end{array}\\ \begin{array}{c} C1\\ C2\\ C3\\ C4 \end{array} & \left[\begin{array}{cccc} 1 & 0.7 & 1.75 & 0.5833\\ 0.3571 & 1 & 1.25 & 0.4167\\ 0.1429 & 0.2 & 1 & 0.1667\\ 0.4286 & 0.6 & 1.5 & 1 \end{array}\right] \end{array}$$

Pairwise comparison results of alternatives with respect to criteria *C*1–*C*2 after the implementation of the perturbation factor *e_ij_* = 0.5:(13)$$\begin{array}{cc} & \begin{array}{cccc} \;\;A1\; & \;\;A2 & \;\;\;A3\; & \;\;\;A4\; \end{array}\\ \begin{array}{c} A1\\ A2\\ A3\\ A4 \end{array} & \left[\begin{array}{cccc} 1 & 0.7 & 1.1667 & 0.7\\ 0.3571 & 1 & 0.8333 & 0.5\\ 0.2143 & 0.3 & 1 & 0.3\\ 0.3571 & 0.5 & 0.8333 & 1 \end{array}\right] \end{array}$$

Pairwise comparison results of alternatives with respect to criteria *C*3–*C*4 after the implementation of the perturbation factor *e_ij_* = 0.5:(14)$$\begin{array}{cc} & \begin{array}{cccc} A1\;\;\;\; & A2\;\;\; & \;\;A3\; & \;\;\;\;A4\;\; \end{array}\\ \begin{array}{c} A1\\ A2\\ A3\\ A4 \end{array} & \left[\begin{array}{cccc} 1 & 0.3333 & 0.1429 & 0.125\\ 0.75 & 1 & 0.2143 & 0.1875\\ 1.75 & 1.1667 & 1 & 0.4375\\ 2 & 1.3333 & 0.5714 & 1 \end{array}\right] \end{array}$$


**Example 2C:**


Pairwise comparison results of criteria with respect to the goal after the implementation of the perturbation factor *e_ij_* = 0.5 and rounding errors (each element of the particular PCM is rounded to Saaty’s scale):(15)$$\begin{array}{cc} & \begin{array}{cccc} \;C1\; & \;\;C2\; & C3\;\; & C4\; \end{array}\\ \begin{array}{c} C1\\ C2\\ C3\\ C4 \end{array} & \left[\begin{array}{cccc} 1 & 0.5 & 2 & 0.5\\ 0.3333 & 1 & 1 & 0.5\\ 0.1429 & 0.2 & 1 & 0.1667\\ 0.5 & 0.5 & 2 & 1 \end{array}\right] \end{array}$$

Pairwise comparison results of alternatives with respect to criteria *C*1–*C*2 after the implementation of the perturbation factor *e_ij_* = 0.5 and rounding errors (each element of the particular PCM is rounded to Saaty’s scale):(16)$$\begin{array}{cc} & \begin{array}{cccc} A1\;\; & \;\;\;A2\; & \;A3 & \;\;A4 \end{array}\\ \begin{array}{c} A1\\ A2\\ A3\\ A4 \end{array} & \left[\begin{array}{cccc} 1 & 0.5 & 1 & 0.5\\ 0.3333 & 1 & 1 & 0.5\\ 0.2 & 0.3333 & 1 & 0.3333\\ 0.3333 & 0.5 & 1 & 1 \end{array}\right] \end{array}$$

Pairwise comparison results of alternatives with respect to criteria *C*3–*C*4 after the implementation of the perturbation factor *e_ij_* = 0.5 and rounding errors (each element of the particular PCM is rounded to Saaty’s scale):(17)$$\begin{array}{cc} & \begin{array}{cccc} A1\;\; & \;\;A2\;\;\; & \;A3\;\; & \;\;\;A4 \end{array}\\ \begin{array}{c} A1\\ A2\\ A3\\ A4 \end{array} & \left[\begin{array}{cccc} 1 & 0.3333 & 0.1429 & 0.125\\ 1 & 1 & 0.2 & 0.2\\ 2 & 1 & 1 & 0.5\\ 2 & 1 & 0.5 & 1 \end{array}\right] \end{array}$$


**Example 2D:**


Pairwise comparison results of criteria with respect to the goal after the implementation of the perturbation factor *e_ij_* = 0.5, rounding errors (each element of the particular PCM is rounded to Saaty’s scale) and forced reciprocity errors (the PCM is transformed to be reciprocal in such a way that only elements from above its diagonal are taken into consideration, and elements below its diagonal are replaced by inverses of their counterparts from above its diagonal), with $PPV_{PREV}^{GOAL}$ computed on the basis of the obtained PCM with the application of PREV:(18)$$\begin{array}{cc} & \begin{array}{cccc} C1 & C2 & C3 & C4 \end{array}\\ \begin{array}{c} C1\\ C2\\ C3\\ C4 \end{array} & \left[\begin{array}{cccc} 1 & 0.5 & 2 & 0.5\\ 2 & 1 & 1 & 0.5\\ 0.5 & 1 & 1 & 0.1667\\ 2 & 2 & 6 & 1 \end{array}\right] \end{array}\begin{array}{cc} & \begin{array}{c} PPV_{PREV}^{GOAL}\\ \left[\begin{array}{c} 0.18215\\ 0.22278\\ 0.12115\\ 0.47392 \end{array}\right] \end{array} \end{array}$$

Pairwise comparison results of alternatives with respect to criteria *C*1–*C*2 after the implementation of the perturbation factor *e_ij_* = 0.5, rounding errors (each element of the particular PCM is rounded to Saaty’s scale) and forced reciprocity errors (the PCM is transformed to be reciprocal in such a way that only elements from above its diagonal are taken into consideration, and elements below its diagonal are replaced by inverses of their counterparts from above its diagonal), with $PPV_{PREV}^{C1C2}$ computed on the basis of the obtained PCM with the application of PREV:(19)$$\begin{array}{cc} & \begin{array}{cccc} A1 & A2 & A3\; & A4\;\;\;\; \end{array}\\ \begin{array}{c} A1\\ A2\\ A3\\ A4 \end{array} & \left[\begin{array}{cccc} 1 & 0.5 & 1 & 0.5\\ 2 & 1 & 1 & 0.5\\ 1 & 1 & 1 & 0.3333\\ 2 & 2 & 3 & 1 \end{array}\right] \end{array}\begin{array}{cc} & \begin{array}{c} PPV_{PREV}^{C1C2}\\ \left[\begin{array}{c} 0.16406\\ 0.23278\\ 0.17510\\ 0.42806 \end{array}\right] \end{array} \end{array}$$

Pairwise comparison results of alternatives with respect to criteria *C*3–*C*4 after the implementation of the perturbation factor *e_ij_* = 0.5, rounding errors (each element of the particular PCM is rounded to Saaty’s scale) and forced reciprocity errors (the PCM is transformed to be reciprocal in such a way that only elements from above its diagonal are taken into consideration, and elements below its diagonal are replaced by inverses of their counterparts from above its diagonal), with $PPV_{PREV}^{C3C4}$ computed on the basis of the obtained PCM with the application of PREV:(20)$$\begin{array}{cc} & \begin{array}{cccc} A1\;\; & A2\;\;\; & \;\;A3\; & \;\;\;\;A4\;\; \end{array}\\ \begin{array}{c} A1\\ A2\\ A3\\ A4 \end{array} & \left[\begin{array}{cccc} 1 & 0.3333 & 0.1429 & 0.125\\ 3 & 1 & 0.2 & 0.2\\ 7 & 5 & 1 & 0.5\\ 8 & 5 & 2 & 1 \end{array}\right] \end{array}\begin{array}{cc} & \begin{array}{c} PPV_{PREV}^{C3C4}\\ \left[\begin{array}{c} 0.04698\\ 0.10012\\ 0.34774\\ 0.50517 \end{array}\right] \end{array} \end{array}$$

After standard AHP synthesis, the following result is obtained: TCPV_PREV_ = [0.09439, 0.15383, 0.27783, 0.47394]^T^, which again is different from HTPV = [0.25, 0.21, 0.23, 0.31]^T^. Comparing HTPV with the estimated TCPV_PREV_, the performance measures mentioned before—i.e., SRCC, PCC and MAD—which reflect the approximation quality of PREV can be established again. For the above exemplary values of HTPV and TCPV_PREV_, these measures are different than before and are as follows: SRCC = 0.4, PCC = 0.76944, MAD = 0.10589. Surprisingly, SRCC in this case is twice as high as in the first example, although this time, more PCM distortions were applied.

## 4. Results of Complete Examination with Discussion

Noticeably, the comparison of the approximation quality of any PDM available for PCB ranking problems is possible as presented in Section 3. It is thus possible to examine a few selected PDMs for the AHP. Thus, the MCS designed for this purpose—i.e., processing the algorithm emulating steps from Examples 2A–2D 25,000 times (250 distinctive AHP frameworks perturbed 100 times each)—provided the correlation scores presented in Table A1, Table A2, Table A3 and Table A4. However, in Table 5, Table 6, Table 7 and Table 8, discrepancies among correlation scores obtained by the examined PDMs are presented in relation to the selected referential values, which in these cases constitute the correlation scores obtained by PREV.

The tables should be read from left to right and row by row from the top to the bottom. In every first three columns on the left side of the tables, simulation parameters taken into consideration are provided; i.e., the applied preference scale, the interval for the perturbation factor, and sets for alternatives and criteria applied during MCS. Then, differences between performance statistics are given for each simulation scenario.

It should be emphasized here that many PDMs have been proposed thus far, and their effectiveness has been evaluated by various means. Various research and different measures of the effectiveness of PDMs lead to different conclusions [97,98]. For example, Choo et al. [99] in their research recommend LLSM as the best PDM with a simple formula for computing PVs, equipped with many desirable properties discovered by Fichtner [100], and the method is very popular as the best alternative for PREV. On the other hand, some support for SNCS also exists—e.g., [36,47,101]—and LSDM is also considered as an efficient PDM; see, e.g., [30]. Basically, it is agreed that many research results, including those of this work, do not provide support for the recommendation by Saaty and Vargas [42] and Saaty and Hu [55] that PREV reputedly is the only PDM which should be used when pairwise comparisons are not entirely consistent. It is also agreed, as stated by Golany and Kress [102], that the selection of a PDM for PCB problems should be dictated by the desired measure of the PDM’s effectiveness, as different error measures support mathematically different PDMs. Hence, as suggested by Bajwa et al. [101], the defining question is not which PDM is superior, but which application results are expected and/or what level of effectiveness or performance criteria are more valued. It is also believed that this research supports the conclusion stated by Saaty and Hu [55] that there is a difference between metric topology and order topology, where in the former the central concern is closeness and in the latter both closeness and order preservation features are equally important. It can be agreed after all these years of research that none of the examined PDMs have been found to be universally superior to all others in all aspects. However, to the best of our knowledge, for the first time, a statistical foundation has been created to evaluate scenarios when PDMs coincide and when their discrepancies are statistically significant. Hence, the possibility was created for a DM to assess the risk of accepting an ineffective PDM or rejecting an effective PDM—the standard problem known to every statistician and very important to each DM during the statistical evaluation of decisional options; i.e., statistical alternative hypothesis testing.

For this research, four distinctive PDM have been selected on the basis of different criteria. PREV is studied because it was conceived with the AHP. LLSM is considered because it has a simplified form and is usually promoted as the best alternative for PREV. SNCS is taken into consideration here for its simplicity and good effectiveness as shown in, e.g., [19,47]. LSDM is examined because it combines spectral theory with an optimization-based approach to PCB problems.

As stated earlier, all the above PDMs have been more or less intensively studied and have shown their effectiveness, efficiency and desired analytical properties [34,103]. They have been evaluated from the perspective of various measures of effectiveness; e.g., Mean Square Error (MSE), Mean Absolute Deviation (MAD), Mean Central Conformity (MCC), Mean Rank Violation (MRV) (see, e.g., [19,36,47,102]), the Coefficient of Multiple Determination (CMD = *R*^2^), which is widely applied in regression analysis (see, e.g., [104]), the Garuti Compatibility Index (GCI) and the Saaty Compatibility Index (SCI) (see, e.g., [58,105,106]). However, in this research, focus was given to statistical measures of examined phenomena and their statistical significance; thus, emphasis was placed on the introduced PDM approximation quality measures; i.e., mostly MSRC, but also MPCC and MAAD. Thus, the discrepancies in the correlation scores presented in Table 5, Table 6, Table 7 and Table 8 are analyzed from the perspective of the PDM rank preservation capability designated by MSRC and the general correlation significance determined by MPCC.

As can be seen, all PDMs perform steadily under the four MCS scenarios presented in Table 5, Table 6, Table 7 and Table 8. It should be noted that PREV does not always outperform the other PDMs in this study; in many cases, it is actually the other way round. The selected PDMs for the examination perform better from the perspective of the approximation quality represented by MSRC. In relation to this phenomenon, it occurs that LSDM dominates over PREV most often in comparison with the other PDMs and from the perspective of all applied MCS scenarios (see the numerical lower subscript of the particular PDM for information on how many times the indexed PDM prevails PREV). This is an important piece of information as the approximation quality of the DM preference intensities of any PDM seems a very crucial issue in PCB problems.

Fortunately, the information provided in Table 5, Table 6, Table 7 and Table 8 can be analyzed from the statistical perspective because the significance of the difference between any two correlation coefficients (CC), denoted as CC[1] and CC[2], can be tested using “*t*” statistics defined by the following formula:(21)$$t=R\sqrt{\left(n-2\right)/\left(1-R^{2}\right)}$$
where *R* is the difference between particular CC.

These statistics have a distribution of *t–student* with *n-2* degrees of freedom *df*, where *n* equals the size of the sample. Thus, the following hypothesizes can be tested:
$$H_{0}:\;\left|\mathrm{CC}\left[1]-\mathrm{CC}[2\right]\right|=0\;\mathrm{versus}\;H_{1}:\;\left|\mathrm{CC}\left[1]-\mathrm{CC}[2\right]\right|>0,$$
and conversely,
$$H_{0}:\;\left|\mathrm{CC}\left[1]-\mathrm{CC}[2\right]\right|>0\;\mathrm{versus}\;H_{1}:\;\left|\mathrm{CC}\left[1]-\mathrm{CC}[2\right]\right|\approx 0.$$

Hence, the following conclusions can be drawn from data provided in Table 5, Table 6, Table 7 and Table 8. If the performance of a particular PDM differs from the performance of PREV by less than 0.00160 (*t* = 0.252972417), then it can be assumed with an 80% confidence level (*t* = 0.253) that its performance discrepancy in relation to the performance of PREV is negligible. For an 85% confidence level (*t* = 0.1895), this discrepancy should be smaller than 0.00118 (*t* = 0.186567049), and for a 90% confidence level (*t* = 0.126), it should be smaller than 0.00076 (*t* = 0.120161779).

On the other hand, if the performance of the particular PDM differs from the performance of PREV by more than 0.00815 (*t* = 1.288619398), then it can be assumed with an 80% confidence level (*t* = 1.282) that its performance discrepancy in relation to the performance of PREV is significant, and this discrepancy should not be neglected. For an 85% confidence level (*t* = 1.452) this discrepancy should be greater than 0.00920 (*t* = 1.454651099), and for a 90% confidence level (*t* = 1.645), it should be greater than 0.01050 (*t* = 1.660220885).

To be able to more completely examine this issue, the MAAD scores for four studied scenarios are presented in Table A5, and discrepancies among the MAAD scores of three PDMs in relation to PREV for four studied scenarios are presented in Table 9.

It can be noticed that all examined PDMs perform quite similarly from the perspective of their approximation quality evaluated by their MAAD. Nevertheless, considering MAAD as the performance criterion, LSDM outperforms PREV 21 times, LLSM is better than PREV 22 times and SNCS outperforms PREV 9 times. In conclusion, as PREV is quite frequently criticized in the literature, this outcome should not be surprising.

Taking into account the above examination results, it was decided to analyze one more scenario with MCS. Generally, the algorithm applied for MCS this time is an expanded version of the approach previously applied and presented in earlier examples provided in this research in Section 4. The algorithm can be specified as follows:*Step 1. Generate a random PV;—i.e., k = [k_1_,..., k_n_]^T^ of size [n x 1]—for the criteria and the associated original unbiased PCM(k) = K(k).**Step 2. Randomly generate exactly n PVs—i.e., a_n_ = [a_n,1_,..., a_n,m_] of size [m x 1]—for the alternatives under each criterion and the associated original unbiased PCMs(a) = A_n_(a).**Step 3. Compute the joint priority vector w of size [m x 1] by the following procedure:**w_x_ = k_1_a_1, x_ + k_2_a_2, x_ +...+ k_n_a_n, x_*(22)*Step 4. Randomly select a number e from the given interval [α,β] based on the given PD.**Step 5. Use step 5A and step 5B separately:**Step 5A is a case of applying forced reciprocity to the PCM:**Replace all elements a_ij_ for i < j of all A_n_(a) with ea_ij_ and all elements k_ij_ for i < j of K(k) with ek_ij_.**Step 5B is the case of the acceptance of nonreciprocal PCM:**Replace all elements a_ij_ for i ≠ j of all A_n_(a) with ea_ij_ and all elements k_ij_ for i ≠ j of K(k) with ek_ij_.**Step 6. Use steps 6A and 6B separately:**Step 6A—if Step 5A is satisfied*,*Round all values of elements a_ij_ for i < j of all A_n_(a) and all values of elements k_ij_ for i < j of K(k) to the nearest values of the scale under consideration, then replace all elements a_ij_ for i > j of all A_n_(a) by 1/a_ij_ and all elements k_ij_ for i > j of K(k) by 1/k_ij_.**Step 6B—after completing step 5B*,*Round all values of elements a_ij_ for i ≠ j of all A_n_(a) and all values of elements k_ij_ for i ≠ j of K(k) to the nearest values of the scale under consideration.**Step 7. Given all perturbed A_n_(a), denoted as A_n_(a)*, and perturbed K(k), denoted as K(k)*, compute their corresponding PVs a_n_* and k* using the given PDM; i.e., PREV, SNCS, LLSM, and LSDM*.*Step 8. Calculate the TPV w*(PDM) of size [m x 1] by the following procedure:**w***_x_* = *k*^*^_1_*a*^*^_1, *x*_ + *k*^*^_2_*a*^*^_2, *x*_ + …+ *k*^*^*_n_ a*^*^*_n_*_, *x*_
(23)*Step 9. Calculate the SRCC for all w*(PDM) and w, as well as any specified quality characteristics of the approximation;—e.g., MAD, PCC, or other relative deviations—e.g., mean relative errors, denoted as*(24)$$MRE_{\gamma ,\chi }\left(w*(PDM),w\right)=\frac{1}{m}\sum_{i=1}^{m}\frac{\left|w_{i}-w_{i}*(PDM)\right|}{w_{i}}$$*or average relative ratios, the value of which is given in* [30] ,* denoted as*
(25)$$MRR_{\gamma ,\chi }\left(w*(PDM),w\right)=\frac{1}{m}\sum_{i=1}^{m}\frac{w_{i}*(PDM)}{w_{i}}$$*Step 10. Repeat steps 4 to 9 χ times, with the sample size denoted asχ.**Step 11. Repeat steps 1 to 9 χ times, with the number of AHP models considered denoted asχ.**Step 12. Return the arithmetic averages of all approximation quality functions computed during all executions in Steps 10 and 11.*

This time, the MCS scenario used was developed with new assumptions in mind. Therefore, not only the results obtained with a reciprocal PCM (RPCM) but also the results obtained with a non-reciprocal PCM (APCM) were considered. Although the AHP does not allow APCM in its structure, it seems reasonable to analyze its application to PCB problems; see, for example, [37,40,71]. It was also decided to introduce new intervals for perturbation factors in MCS and to apply their new PDs. Obviously, this time, the expected value of *e_ij_* was also close to unity; i.e., the value of EV(*e*) ≈ 1. Although this requirement is relatively easy to fulfill based on an asymmetric interval for *e_ij_* (relative to its effect on a particular PCM element), it is quite difficult to realize this assumption with a symmetric interval for *e_ij_*. However, this goal was achieved in this seminal study. It is reasonable to apply symmetric intervals to MCS for *e_ij_* as well, as they more realistically reflect real human performance in pairwise comparisons without strong outliers. Thus, experiments were conducted with different types of PDs, and it was found that the Fisher–Snedecor PD has a property that may be useful for the intended purpose. Namely, for *n_1_* = 14 and *n_2_* = 40 degrees of freedom for 1000 randomly generated numbers based on this PD, their mean is 1.03617, meaning that it is very close to one, and these numbers range from 0.174526 to 5.57826. Under these assumptions, *e*∈[0.174526, 5.57826] therefore holds, giving a completely symmetric PD for *e_ij_*; i.e., EV(*e*) ≈ 1. The results for the selected PDMs and their assumed performance quality measures—i.e., mean relative error (MSRC), mean relative error (MARE), and mean relative ratio (MARR)—derived from the MCS scenario described earlier are shown in Table 10.

As can be seen again, PREV is not the dominant PDM in terms of all simulation scenarios in the established framework (it ranks second overall ex aequo with LLSM). Of course, the obvious differences in the quality of the PV approximation depending on the chosen PDMs are evident for non-reciprocal PDMs. LSDM and LLSM outperform the other chosen PDMs, especially with respect to rank correlations, which are crucial for the phenomenon of rank preservation.

The issue of the rank reversal phenomenon (conservation of preference power) was introduced by Bana e Costa and Vansnick [13]. They gave the following definition: for all alternatives A1, A2, A3 and A4 such that A1 dominates A2 and A3 dominates A4 and the degree of dominance of A1 over A2 is greater than the degree of dominance of A3 over A4, not only w1 > w2 and w3 > w4, but also w1/w2 > w3/w4 is true for the obtained PV. Therefore, the following scenario considered by Bana e Costa and Vansnick is revisited.

When the PCM is given as
$$\left[\begin{array}{ccccc} 1 & 2 & 3 & 5 & 9\\ 1/2 & 1 & 2 & 4 & 9\\ 1/3 & 1/2 & 1 & 2 & 8\\ 1/5 & 1/4 & 1/2 & 1 & 7\\ 1/9 & 1/9 & 1/8 & 1/7 & 1 \end{array}\right]$$
according to the common linguistic interpretation of the AHP, A1 is *strongly* dominant over A4 (A1/A4 = 5) and A4 is *very strongly* dominant over A5 (A4/A5 = 7). This implies that A1/A4 < A4/A5. 

However, the PV obtained from PREV gives [0.4262, 0.2809, 0.1652, 0.1008, 0.0269]^T^ and gives A1/A4 = 4.218 > A4/A5 = 3.741, which violates the condition of order preservation (COP). On the other hand, the PV obtained, e.g., by LSDM gives [0.434659, 0.282449, 0.163602, 0.097671, 0,021620]^T^ and results in ratios if A1/A4 = 4.450245 < A4/A5 = 4.517668, which, unlike PREV, satisfy the COP!

This phenomenon of LSDM is especially interesting because of the perfect rank correlations for PVs derived by LSDM and PREV from randomly generated (uniform probability distribution) transitive and reciprocal inconsistent PCMs (TRPCM) (see Table 11).

In order to compare the results obtained using LSDM with those obtained using PREV and to see if they are the same or if there is a possibility to reverse the order between their PV elements, 1000 TRPCMs were generated using MCS. For each randomly generated TRPCM, PVs were determined: PV_LSDM_ and PV_PREV_ were calculated using LSDM and PREV, respectively. In addition, the PCCs between the PV elements and the SRCCs between their priorities were calculated. The numbers of alternatives *n* considered were chosen as follows: 3, 4, 5, 6, 7, 8, 9, 10 and 12. The number of criteria was set to one. A standard numerical AHP scale was used to express the judgment; i.e., integers 1–9 and their inversions.

Table 11 shows the mean correlation coefficients between PV elements and the priority ranks obtained during MCS with respect to the number of alternatives taken into consideration.

Considering the results presented in Table 11, three facts can be noted: firstly, for *n* = 3, both methods perfectly coincide; secondly, MSRC values for *n* > 3 equal 1,meaning that there is no rank reversal phenomenon between the LSDM and PREV for 1000 randomly generated TRPCMs; lastly, MPCC values for *n* > 3 between PVs derived by the LSDM and PREV for 1000 TRPCMs practically coincide with unity—i.e., MPCC ≈ 1—which indicates the almost perfect coincidence of both PDMs.

## 5. Conclusions

Discrepancies and similarities among examined PDMs have been examined in this research paper from various perspectives, also including the statistical approach. For this purpose, selected statistical measures of the effectiveness of PDMs (approximation quality) have been applied in this research; i.e., MSRC, MPCC, MARE, MAAD and MARR. Information concerning the statistical significance of discrepancies and similarities among examined PDMs is clearly presented. In this way, to the best of our knowledge, for the first time, a statistical foundation has been created to identify situations in which PDMs coincide and their discrepancies can be considered as negligible, and when their discrepancies are statistically significant and they should not be neglected. Hence, the possibility was created for a DM to assess the risk of accepting an ineffective PDM or rejecting an effective PDM—the standard problem known to every statistician and which is very important to each DM during the statistical evaluation of decisional options; i.e., statistical alternative hypothesis testing. The ranking of the PDMs evaluated in the manuscript based on novel scenarios of Monte Carlo simulations was also presented in this research paper. Furthermore, a certain interesting advantage of LSDM which other evaluated PDMs do not have—i.e., the condition of order preservation satisfaction—is also presented in the article. These research accomplishments will certainly provide fundamental support for any DM during their decisions regarding which PDM to choose in various circumstances. 

Given the reality of our physical world, no study is perfect. In order to compare the characteristics of the estimates obtained in the simulation process for the selected PDM, different situations related to various sources of the PCM inconsistency were simulated. Fundamentally, PCM inconsistency commonly results from errors caused by the nature of human judgments and errors due to the technical realization of the pairwise comparison procedure; i.e., rounding errors and errors resulting from the forced reciprocity requirement commonly imposed in PCB ranking problems. All the above errors can be simulated, but the nature of human judgments is represented here as the realization of a stochastic process in accordance with the assumed probability distribution of the perturbation factor; e.g., uniform, gamma, truncated normal and log-normal. As this is only a process generated by a computer, it represents a certain limitation of the presented research. Thus, there is a space for further research in this area with the application of different MCS scenarios, various other measures of PDM effectiveness (performance quality) and a case-based methodology.

## Figures and Tables

**Table 1 entropy-23-01150-t001:** Formulae for the examined PDM.

PDM Name	PDM Formula
Logarithmic Least Squares Method—LLSM	$w_{i\left(LLSM\right)}={\left(\prod_{j=1}^{n}a_{ij}\right)}^{1/n}/{\sum_{i=1}^{n}\left(\prod_{j=1}^{n}a_{ij}\right)}^{1/n}$
Simple Normalized Column Sum—SNCS	$w_{i}{}_{\left(SNCS\right)}=\frac{1}{n}\sum_{j=1}^{n}\left(a_{ij}/\sum_{k=1}^{n}a_{kj}\right)$
Logarithmic Squared Deviations Minimization Method—LSDM	$w_{\left(LSDM\right)}=\min\sum_{i=1}^{n}\ln^{2}\left(\sum_{j=1}^{n}\left(a_{ij}w_{j}/nw_{i}\right)\right)$

**Table 2 entropy-23-01150-t002:** Formulae for the performance measures.

Performance Measures Names	Performance Measures Formulae
Mean Spearman Ranks Correlation Coefficient	$MSRC=\frac{1}{N}{\sum_{t=1}^{N}\left(1-6\sum_{i=1}^{n}d_{i}^{2}/n\left(n^{2}-1\right)\right)}_{t}$
Mean Pearson Correlation Coefficient	$MPCC=\frac{1}{N}{\sum_{t=1}^{N}\left(\frac{\sum_{i=1}^{n}\left(w_{i}-\bar{w}\right)\left(v_{i}-\bar{v}\right)}{\sqrt{\sum_{i=1}^{n}{\left(w_{i}-\bar{w}\right)}^{2}\sum_{i=1}^{n}{\left(v_{i}-\bar{v}\right)}^{2}}}\right)}_{t}$
Mean Average Absolute Deviation	$MAAD=\frac{1}{N}{\sum_{t=1}^{N}\left(\frac{1}{n}\sum_{i=1}^{n}\left|w_{i}-v_{i}\right|\right)}_{t}$

*d_i_*—difference between the two ranks of the considered PVs’ respective elements, *n*—the number of examined elements within a single experiment, *N*—the number of experiment iterations; *w_i_*, *v_i_—i-th* elements of the respective PVs that are compared.

**Table 3 entropy-23-01150-t003:** Approximation quality of four PDMs for 10,000 iterations of the AHP random framework^#^ with the application of rounding errors and reciprocity imposition errors.

PDM Name	MSRC	MPCC	MAAD
LLSM	0.962429	0.997569	0.01083350
PREV	0.962881	0.998005	0.00994892
SNCS	0.961548	0.997740	0.01095650
LSDM	0.963619	0.997964	0.01008080

^#^ a random framework represents a uniformly drawn number of criteria (*n_k_*) and number of alternatives (*n_a_*) in the single AHP model; in this scenario, $n_{k},\;n_{a}\in \left\{5,\;6,\;7,\;8,\;9\right\}$.

**Table 4 entropy-23-01150-t004:** Approximation quality of four PDMs for 10,000 iterations of the AHP random framework ^#^, i.e., $n_{k},\;n_{a}\in \left\{5,\;6\ldots ,15\right\}$, with the application of rounding errors and reciprocity imposition errors.

PDM Name	MSRC	MPCC	MAAD
LLSM	0.972270	0.996995	0.00806526
PREV	0.972379	0.997704	0.00724835
SNCS	0.971815	0.997441	0.00788974
LSDM	0.972510	0.997620	0.00736441

^#^ a random framework represents a uniformly drawn number of criteria (*n_k_*) and number of alternatives (*n_a_*) in the single AHP model.

**Table 5 entropy-23-01150-t005:** Absolute discrepancies of the performance of arbitrarily selected PDMs and PREV for 25,000 cases of various uniformly drawn and uniformly perturbed AHP frameworks ^(%)^.

Simulation Parameters			^($)^ STAT	PREV	LLSM	SNCS	LSDM
Saaty’s Scale	*e_ij_*∈[0.75,1.25]	*n_k_*,*n_a_*∈{3,4…,7}	MSRC MPCC	Referential value located in Table A1.	0.000160	0.000280	0.000080
					0.000180	0.000246	0.000017
		*n_k_*,*n_a_*∈{8,9…,12}	MSRC MPCC		0.000211	0.000787	0.000091
					0.000608	0.000138	0.000073
	*e_ij_*∈[0.05,1.95]	*n_k_*,*n_a_*∈{3,4…,7}	MSRC MPCC		0.014305	0.009523	0.004557
					0.006454	0.007787	0.001434
		*n_k_*,*n_a_*∈{8,9…,12}	MSRC MPCC		0.027007	0.014009	0.006119
					0.016304	0.011317	0.002445
Geometric Scale	*e_ij_*∈[0.75,1.25]	*n_k_*,*n_a_*∈{3,4…,7}	MSRC MPCC		0.001754	0.000463	0.000003
					0.000263	0.000281	0.000036
		*n_k_*,*n_a_*∈{8,9…,12}	MSRC MPCC		0.000880	0.001085	0.000014
					0.000380	0.000190	0.000048
	*e_ij_*∈[0.05,1.95]	n_k_,n_a_∈{3,4…,7}	MSRC MPCC		0.011820	0.012460	0.003883
					0.010472	0.011425	0.002947
		*n_k_*,*n_a_*∈{8,9…,12}	MSRC MPCC		0.048165	0.026215	0.010913
					0.033336	0.023230	0.007889
Selected Exemplary Scale	*e_ij_*∈[0.75,1.25]	*n_k_*,*n_a_*∈{3,4…,7}	MSRC MPCC		0.000345	0.000160	0.000088
					0.000021	0.000002	0.000001
		*n_k_*,*n_a_*∈{8,9…,12}	MSRC MPCC		0.000317	0.000028	0.000061
					0.000043	0.000037	0.000010
	*e_ij_*∈[0.05,1.95]	*n_k_*,*n_a_*∈{3,4…,7}	MSRC MPCC		0.015625	0.009485	0.004011
					0.011447	0.011942	0.003528
		*n_k_*,*n_a_*∈{8,9…,12}	MSRC MPCC		0.048393	0.029171	0.011378
					0.034861	0.026269	0.008804

^(%)^ All simulation scenarios imposed reciprocity conditions for every examined PCM within each AHP framework, uniformly drawn perturbation factors from the indicated interval and rounding errors connected with the assigned preference scale. The scenario assumed 100 perturbations of 250 distinctive AHP frameworks. ^($)^ STAT stands for “statistics”. The selected exemplary scale was a simple ordinal scale from 1 to 50.

**Table 6 entropy-23-01150-t006:** Absolute discrepancies of the performance of arbitrarily selected PDMs and PREV for 25,000 cases of various uniformly drawn and log-normally perturbed AHP frameworks ^(%)^.

Simulation Parameters			^($)^ STAT	PREV	LLSM	SNCS	LSDM
Saaty’s Scale	*e_ij_*∈[0.75,1.25]	*n_k_*,*n_a_*∈{3,4…,7}	MSRC MPCC	Referential value located in Table A2.	0.000209	0.000214	0.000103
					0.000497	0.000565	0.000080
		*n_k_*,*n_a_*∈{8,9…,12}	MSRC MPCC		0.001472	0.001332	0.000119
					0.000827	0.000330	0.000130
	*e_ij_*∈[0.05,1.95]	*n_k_*,*n_a_*∈{3,4…,7}	MSRC MPCC		0.003203	0.000480	0.002054
					0.002449	0.001016	0.001510
		*n_k_*,*n_a_*∈{8,9…,12}	MSRC MPCC		0.003215	0.001696	0.001995
					0.002923	0.001597	0.002058
Geometric Scale	*e_ij_*∈[0.75,1.25]	*n_k_*,*n_a_*∈{3,4…,7}	MSRC MPCC		0.001266	0.001071	0.000189
					0.000069	0.000121	0.000009
		*n_k_*,*n_a_*∈{8,9…,12}	MSRC MPCC		0.000476	0.000335	0.000105
					0.000428	0.000267	0.000050
	*e_ij_*∈[0.05,1.95]	*n_k_*,*n_a_*∈{3,4…,7}	MSRC MPCC		0.005631	0.001977	0.002068
					0.002043	0.001931	0.001307
		*n_k_*,*n_a_*∈{8,9…,12}	MSRC MPCC		0.009619	0.005352	0.004868
					0.005899	0.001343	0.003321
Selected Exemplary Scale	*e_ij_*∈[0.75,1.25]	*n_k_*,*n_a_*∈{3,4…,7}	MSRC MPCC		0.000523	0.000129	0.000075
					0.000036	0.000170	0.000002
		*n_k_*,*n_a_*∈{8,9…,12}	MSRC MPCC		0.001390	0.000342	0.000209
					0.000013	0.000082	0.000007
	*e_ij_*∈[0.05,1.95]	*n_k_*,*n_a_*∈{3,4…,7}	MSRC MPCC		0.003425	0.000914	0.002662
					0.002411	0.001400	0.001723
		*n_k_*,*n_a_*∈{8,9…,12}	MSRC MPCC		0.003060	0.000279	0.003384
					0.004000	0.001726	0.002667

^(%)^ All simulation scenarios imposed reciprocity conditions for every examined PCM within each AHP framework, log-normally drawn perturbation factors from the indicated interval and rounding errors connected with the assigned preference scale. The scenario assumed 100 perturbations of 250 distinctive AHP frameworks. ^($)^ STAT stands for “statistics”. The selected exemplary scale was a simple ordinal scale from 1 to 50.

**Table 7 entropy-23-01150-t007:** Absolute discrepancies in the performance of arbitrarily selected PDMs and PREV for 25,000 cases of various uniformly drawn and truncated-normally perturbed AHP frameworks ^(%)^.

Simulation Parameters			^($)^ STAT	PREV	LLSM	SNCS	LSDM
Saaty’s Scale	*e_ij_*∈[0.75,1.25]	*n_k_*,*n_a_*∈{3,4…,7}	MSRC MPCC	Referential value located in Table A3.	0.002020	0.004480	0.000311
					0.000368	0.000523	0.000051
		*n_k_*,*n_a_*∈{8,9…,12}	MSRC MPCC		0.000415	0.000820	0.000037
					0.000494	0.000219	0.000052
	*e_ij_*∈[0.05,1.95]	*n_k_*,*n_a_*∈{3,4…,7}	MSRC MPCC		0.003180	0.002840	0.000317
					0.000239	0.000165	0.000034
		*n_k_*,*n_a_*∈{8,9…,12}	MSRC MPCC		0.005738	0.002650	0.000702
					0.000411	0.000384	0.000004
Geometric Scale	*e_ij_*∈[0.75,1.25]	*n_k_*,*n_a_*∈{3,4…,7}	MSRC MPCC		0.000222	0.001437	0.000029
					0.000242	0.000518	0.000032
		*n_k_*,*n_a_*∈{8,9…,12}	MSRC MPCC		0.000762	0.000912	0.000038
					0.000292	0.000126	0.000022
	*e_ij_*∈[0.05,1.95]	*n_k_*,*n_a_*∈{3,4…,7}	MSRC MPCC		0.002917	0.001189	0.000346
					0.001056	0.000693	0.000191
		*n_k_*,*n_a_*∈{8,9…,12}	MSRC MPCC		0.008026	0.004473	0.000812
					0.001274	0.000787	0.000164
Selected Exemplary Scale	*e_ij_*∈[0.75,1.25]	*n_k_*,*n_a_*∈{3,4…,7}	MSRC MPCC		0.001392	0.000851	0.000132
					0.000064	0.000059	0.000004
		*n_k_*,*n_a_*∈{8,9…,12}	MSRC MPCC		0.000032	0.000095	0.000007
					0.000047	0.000033	0.000011
	*e_ij_*∈[0.05,1.95]	*n_k_*,*n_a_*∈{3,4…,7}	MSRC MPCC		0.004248	0.001957	0.000957
					0.001598	0.001334	0.000270
		*n_k_*,*n_a_*∈{8,9…,12}	MSRC MPCC		0.009329	0.004291	0.001371
					0.001861	0.001278	0.000227

^(%)^ All simulation scenarios imposed reciprocity conditions for every examined PCM within each AHP framework, truncated-normally drawn perturbation factors from the indicated interval and rounding errors connected with the assigned preference scale. The scenario assumed 100 perturbations of 250 distinctive AHP frameworks. ^($)^ STAT stands for “statistics”. The selected exemplary scale was a simple ordinal scale from 1 to 50.

**Table 8 entropy-23-01150-t008:** Absolute discrepancies in the performance of arbitrarily selected PDMs and PREV for 25,000 cases of various uniformly drawn and gamma perturbed AHP frameworks ^(%)^.

Simulation Parameters			^($)^ STAT	PREV	LLSM	SNCS	LSDM
Saaty’s Scale	*e_ij_*∈[0.75,1.25]	*n_k_*,*n_a_*∈{3,4…,7}	MSRC MPCC	Referential value located in Table A4.	0.001517	0.000143	0.000011
					0.000384	0.000380	0.000050
		*n_k_*,*n_a_*∈{8,9…,12}	MSRC MPCC		0.000330	0.001173	0.000027
					0.000599	0.000243	0.000092
	*e_ij_*∈[0.05,1.95]	*n_k_*,*n_a_*∈{3,4…,7}	MSRC MPCC		0.015957	0.013069	0.003780
					0.009327	0.014993	0.002470
		*n_k_*,*n_a_*∈{8,9…,12}	MSRC MPCC		0.027611	0.021041	0.004950
					0.024458	0.025688	0.001128
Geometric Scale	*e_ij_*∈[0.75,1.25]	*n_k_*,*n_a_*∈{3,4…,7}	MSRC MPCC		0.000600	0.000137	0.000294
					0.000242	0.000263	0.000028
		*n_k_*,*n_a_*∈{8,9…,12}	MSRC MPCC		0.000543	0.000326	0.000154
					0.000322	0.000251	0.000046
	*e_ij_*∈[0.05,1.95]	*n_k_*,*n_a_*∈{3,4…,7}	MSRC MPCC		0.019162	0.023100	0.007222
					0.019843	0.030935	0.006085
		*n_k_*,*n_a_*∈{8,9…,12}	MSRC MPCC		0.048069	0.034347	0.013898
					0.057891	0.047235	0.012189
Selected Exemplary Scale	*e_ij_*∈[0.75,1.25]	*n_k_*,*n_a_*∈{3,4…,7}	MSRC MPCC		0.000123	0.001300	0.000066
					0.000043	0.000029	0.000006
		*n_k_*,*n_a_*∈{8,9…,12}	MSRC MPCC		0.000518	0.000340	0.000061
					0.000011	0.000108	0.000007
	*e_ij_*∈[0.05,1.95]	*n_k_*,*n_a_*∈{3,4…,7}	MSRC MPCC		0.033649	0.037263	0.015775
					0.035832	0.051840	0.014031
		*n_k_*,*n_a_*∈{8,9…,12}	MSRC MPCC		0.079560	0.062324	0.032245
					0.124541	0.108218	0.041822

^(%)^ All simulation scenarios imposed reciprocity conditions for every examined PCM within each AHP framework, gamma drawn perturbation factors from the indicated interval and rounding errors connected with the assigned preference scale. The scenario assumed 100 perturbations of 250 distinctive AHP frameworks. ^($)^ STAT stands for “statistics”. The selected exemplary scale was a simple ordinal scale from 1 to 50.

**Table 9 entropy-23-01150-t009:** Performance discrepancies of three arbitrarily selected PDMs in relation to PREV for 25,000 cases of different uniformly drawn and variously perturbed AHP frameworks ^(%)^.

Simulation Parameters			^($)^ PDM	MAAD^[1]^	MAAD^[2]^	MAAD^[3]^	MAAD^[4]^
Saaty’s Scale	*e_ij_*∈[0.75,1.25]	*n_k_*,*n_a_*∈{3,4…,7}	PREV	Referential value located in Table A5.			
			LLSM SNCS LSDM	0.000584	0.000545	0.000719	0.000493
				0.001399	0.001240	0.001820	0.001487
				0.000077	0.000094	0.000115	0.000086
		*n_k_*,*n_a_*∈{8,9…,12}	PREV	Referential value located in Table A5.			
			LLSM SNCS LSDM	0.000703	0.000721	0.000744	0.000536
				0.000503	0.000765	0.000551	0.000630
				0.000109	0.000140	0.000097	0.000094
	*e_ij_*∈[0.05,1.95]	*n_k_*,*n_a_*∈{3,4…,7}	PREV	Referential value located in Table A5.			
			LLSM SNCS LSDM	−0.001188	−0.000620	0.000245	−0.001762
				0.001184	0.002267	0.001847	0.000834
				−0.000255	−0.000285	0.000051	−0.000517
		*n_k_*,*n_a_*∈{8,9…,12}	PREV	Referential value located in Table A5.			
			LLSM SNCS LSDM	−0.001394	−0.000319	0.000197	−0.001295
				−0.000133	0.000684	0.000572	−0.001061
				−0.000177	−0.000169	0.000072	−0.000101
Geometric Scale	*e_ij_*∈[0.75,1.25]	*n_k_*,*n_a_*∈{3,4…,7}	PREV	Referential value located in Table A5.			
			LLSM SNCS LSDM	0.000791	0.000366	0.000531	0.000297
				0.001600	0.001125	0.001102	0.001088
				0.000145	0.000064	0.000075	0.000041
		*n_k_*,*n_a_*∈{8,9…,12}	PREV	Referential value located in Table A5.			
			LLSM SNCS LSDM	0.000546	0.000615	0.000637	0.000473
				0.000450	0.000700	0.000404	0.000556
				0.000079	0.000097	0.000074	0.000078
	*e_ij_*∈[0.05,1.95]	*n_k_*,*n_a_*∈{3,4…,7}	PREV	Referential value located in Table A5.			
			LLSM SNCS LSDM	−0.001980	−0.000334	−0.000017	−0.002306
				0.000588	0.002569	0.001050	−0.000887
				−0.000514	−0.000213	0.000008	−0.000792
		*n_k_*,*n_a_*∈{8,9…,12}	PREV	Referential value located in Table A5.			
			LLSM SNCS LSDM	−0.002642	−0.000409	0.000015	−0.002901
				−0.001178	0.000663	0.000431	−0.002264
				−0.000529	−0.000229	0.000037	−0.000691
Selected Exemplary Scale	*e_ij_*∈[0.75,1.25]	*n_k_*,*n_a_*∈{3,4…,7}	PREV	Referential value located in Table A5.			
			LLSM SNCS LSDM	0.000001	0.000009	0.000079	−0.000058
				0.000259	0.000550	0.000325	0.000351
				0.000004	0.000000	0.000018	−0.000006
		*n_k_*,*n_a_*∈{8,9…,12}	PREV	Referential value located in Table A5.			
			LLSM SNCS LSDM	0.000075	0.000063	0.000089	0.000012
				0.000192	0.000347	0.000159	0.000301
				0.000020	0.000025	0.000029	0.000016
	*e_ij_*∈[0.05,1.95]	*n_k_*,*n_a_*∈{3,4…,7}	PREV	Referential value located in Table A5.			
			LLSM SNCS LSDM	−0.002301	−0.000441	−0.000387	−0.004033
				−0.000195	0.002836	0.000351	−0.003573
				−0.000586	−0.000246	−0.000044	−0.001506
		*n_k_*,*n_a_*∈{8,9…,12}	PREV	Referential value located in Table A5.			
			LLSM SNCS LSDM	−0.003147	−0.000283	−0.000486	−0.005799
				−0.001707	0.000858	0.000061	−0.005315
				−0.000677	−0.000199	−0.000044	−0.002030

^(%)^ All simulation scenarios imposed reciprocity conditions for every examined PCM within each AHP framework, uniform MAAD^[1]^, log-normal MAAD^[2]^, truncated-normal MAAD^[3]^, or gamma MAAD^[4]^, with drawn perturbation factors from the indicated interval and rounding errors connected with the assigned preference scale. A negative value of the particular MAAD for PDM indicates that it is smaller than the MAAD of PREV. ^($)^ PDM stands for “priorities deriving method”.

**Table 10 entropy-23-01150-t010:** The average performance of the five selected named PDMs for various uniform constructions of 100,000 AHP models—1000 hypothetical decision problems perturbed 100 times each (*).

Scenario Details			PDM	MARE	Ranks	MSRC	Ranks	MARR	Ranks	∑Ranks.
*Geometric* Scale	*n, m*∈{3, 4…, 7}	RPCM	LLSM	0.123288	3	0.916281	1	1.04646	2	6
			PREV	0.123030	1	0.915056	4	1.04546	1	6
			LSDM	0.123044	2	0.915489	2	1.04699	3	7
			SNCS	0.132926	4	0.915228	3	1.05865	4	11
		APCM	LLSM	0.100511	1	0.930242	3	1.02953	3	7
			PREV	0.101523	3	0.930164	4	1.02938	2	9
			LSDM	0.100658	2	0.930965	2	1.02926	1	5
			SNCS	0.108689	4	0.931026	1	1.04315	4	9
	*n, m*∈{8, 9…, 12}	RPCM	LLSM	0.079748	3	0.931396	1	1.03319	3	7
			PREV	0.079110	1	0.928266	4	1.03116	1	6
			LSDM	0.079321	2	0.928817	2	1.03173	2	6
			SNCS	0.086223	4	0.928799	3	1.03935	4	11
		APCM	LLSM	0.063936	3	0.943393	2	1.02252	3	8
			PREV	0.062735	2	0.942399	4	1.02070	1	7
			LSDM	0.061757	1	0.944593	1	1.02109	2	4
			SNCS	0.068981	4	0.942764	3	1.02879	4	11
*Saaty’s* scale	*n, m*∈{3, 4…, 7}	RPCM	LLSM	0.143650	3	0.911381	1	1.06578	3	7
			PREV	0.142967	1	0.911151	3	1.06498	1	5
			LSDM	0.143069	2	0.911347	2	1.06520	2	6
			SNCS	0.155694	4	0.910735	4	1.07850	4	12
		APCM	LLSM	0.116095	1	0.927455	1	1.04681	2	4
			PREV	0.116994	3	0.926955	4	1.04705	3	10
			LSDM	0.116337	2	0.927129	3	1.04657	1	6
			SNCS	0.127154	4	0.927397	2	1.06051	4	10
	*n, m*∈{8, 9…, 12}	RPCM	LLSM	0.100279	3	0.917231	1	1.04856	3	7
			PREV	0.098084	1	0.915833	3	1.04630	1	5
			LSDM	0.098648	2	0.916245	2	1.04695	2	6
			SNCS	0.106674	4	0.915633	4	1.05424	4	12
		APCM	LLSM	0.078464	3	0.938192	2	1.03563	3	8
			PREV	0.077002	2	0.937837	3	1.03422	1	6
			LSDM	0.076762	1	0.939669	1	1.03469	2	4
			SNCS	0.084307	4	0.937796	4	1.04125	4	12
Total Ranks Sum			LLSM	54	PREV	54	LSDM	44	SNCS	89
Score			2		2		1		3	

Note: (*) AHP models constructed randomly (uniformly) for a given set of criteria and alternatives. The scenario assumes both factors: a disturbance factor obtained with the F-Snedecor probability for *n*_1_ = 14 and *n*_2_ = 40 degrees of freedom and round-off errors associated with a given scale (geometric or Saaty’s). Includes calculation of performance measures for reciprocal PCM (RPCM) or non-reciprocal PCM (APCM).

**Table 11 entropy-23-01150-t011:** Results of comparative studies concerning LSDM and PREV for 1000 RTPCMs.

^($)^ STAT	Number of alternatives (*n*)				
	3	4	5	6	7
MPCC	1	0.999999	0.999997	0.999993	0.999991
MSRC	1	1	1	1	1
^($)^ STAT	Number of alternatives (*n*)				
	8	9	10	11	12
MPCC	0.999989	0.999991	0.999987	0.999997	0.999988
MSRC	1	1	1	1	1

^($)^ STAT stands for “statistics”.

## Data Availability

Datasets were generated in Wolfram’s Mathematica Software 11 concurrently during the study. All statistics obtained during their analysis are an integral part of this research paper.

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
