# Peer review of "On the Statistical Discrepancy and Affinity of Priority Vector Heuristics in Pairwise-Comparison-Based Methods"

_entropy, 2021, doi:10.3390/e23091150_

Round 1

Reviewer 1 Report

I read this new version of the paper in light of the replies received by the author. I regret to write that I still hold the same original doubts on the significance of the results. I’ll try to comment on the replies with an eye to the manuscript.

In his replies the author writes about personal encounters and episodes, but I do not see how this helps answer my doubts on the relevance of the paper. The author write that “That is why my current research paper deals mainly with statistical affinities i.e. similarities among various PDMs, and not their relevant differences”. However, what are similarities if not lack of differences?
My point still stands. There are various methods, and no relevant differences have been found. So, as a user of these methods, I wonder what the contribution of this paper is and how it could help me choose among methods.
The author claims that “I think that we owe DMs information about performance of various PDMs”, but I maintain that there is already a large body of evidence on the similarities between these methods and therefore this manuscript does not bring new significant information. See, for instance:
* Ishizaka, A., & Lusti, M. (2006). How to derive priorities in AHP: a comparative study. Central European Journal of Operations Research14(4), 387-400.
* Choo, E. U., & Wedley, W. C. (2004). A common framework for deriving preference values from pairwise comparison matrices. Computers & operations research31(6), 893-908.
where it is already written that differences between methods are negligible.

Even reading the conclusions of the paper I fail to grasp its contribution. The conclusion section seems more a brief discussion of what was done by others.

The author claims that “I don’t think we should be censors and decide which information is relevant and which is not for DMs”, but as a reviewer I am asked to judge the relevance of contents for the readership.

A marginal note: my comment 3) was answered with “All theorems and definitions are presented at the beginning now.”. The author did not understand that a definition (D3) shall always come before a theorem (T1) recalling it. Also, I repeat: it if a matrix is consistent, then it must be reciprocal and it is therefore redundant to spell out that it is reciprocal.

Reviewer 2 Report

The following few queries should be answered in this paper. 

  1. On line 264: is mentioned That “ This model assumes that relative ratios of some physical attributes of certain objects are predetermined”. But which methodology is selected for that is not mentioned . So, required to Justifying it.
  2. On line 333 : mentioned that 1000-time iterations have been taken So, why this 1000 times no has been selected ? Do you feel that this no is reliable for processing the algorithm. Could you minimize this 1000 steps of iteration?  
  3. Why only Logarithmic Least Squares Method LLSM , Simple Normalized Column Sum SNCS , Logarithmic Squared Deviations Minimization Method LSDM and PREV methods of PDM’s are selected from the rest of methods? Why not others method are selected ?
  4. Logarithmic Squared Deviations Minimization Method LSDM is useful to get more discrimination among alternatives, but its fail when ???=0 Hence due to this drawback, this method is not more applicable and popular in MCDM decision-making. Then also Why you selected this method?
  5. On line 364, the pairwise comparison results of CRITERIA with respect to GOAL are taken in subjective forms. So due to subjective preference values do we consider that this table is generalised table ?
  6. Uncertainty is the conflict situation which occurs due to incomplete information and knowledge of scenario. Uncertainty due to DMs subjective judgement or possibilities or due to the inconsistency or granularity of information. So, in this model Uncertainty is not considered.
  7. Due to uncertainty, imprecise data, cognitive limitations, and incomplete knowledge of system, he/she may or may not be 100 % confident about their score/decision was given. Thus, how much confident DM is about his/her decision/judgement is not reflected in this research.
  8. This is a very important and hot issue and from my point of view the least studied in MCDM literature. Sensitivity analysis(SA) means to learn how the output varies when the inputs change . Hence Validation and robustness of the Results in this paper by changing number of criteria is not carried out. So, feasibility of solution is not verified.
  9. Author has used Mean Spearman Rank Correlation Coefficient and Pearson Rank Correlation Coefficient methods. But ranking of the alternatives will be changed due to change in number of criteria. Then These Rank Correlation Coefficient methods may give different answers. That should be verified in this paper to check the reliability of results.
  10. AHP can’t be able to handle dependent criteria. i.e. All criteria should be mutually exhaustive and exclusive (independent). So, there is no any provision mentioned to handled Dependent Criteria.  
  11. As per my knowledge, present MCDM methods do not represent the reality. It is universally true that no method and nobody can assert that a right and reliable result has been attained. But we need modelling a real scenario trying to replicate as close as possible all its features, and this is not done actually. As per my perception To get precise results is based on a) Assuming that the selected method is mathematical sound, and by considering that most of all scenario requirements, characteristics and issues have been plugged in the mathematical modelling.

Reviewer 3 Report

Dear Author,

Your manuscript provides the results of the comparative analysis of the selected priorities deriving methods (Principal Right Eigenvalue Method - PREV, Logarithmic Least Squared Method - LLSM, Simple Normalized Column Sum Procedure - SNCS and Logarithmic Squared Deviations Minimization Method - LSDM) within AHP (Analytic Hierarchy Process) according to several selected statistical measures (MSRC, MPCC, MAD, MAAD, PCC, MARE, MARR). The statistical analysis is based on the results obtained by Monte Carlo simulations.
There are a lot of previously published papers provided in the references.  
The main issue of this manuscript is that after careful reading there is still a question whether this manuscript is inside of the scope of this journal. It is even emphasized by the fact that among more than 110 references there is not a single reference from this journal!?

There are also several other issues which should be improved to further the quality of the manuscript. These are the following:

  • Avoid bulk reference citations (e.g. [9-41], [91-104] etc.). Instead provide at least some characteristic for each of the references to prove that this reference is necessary to be cited.
  • Since the author (P.3 L.109-111) wrote "As this is only a stochastic process generated by computer, it constitutes a certain limitation on the presented research." it seems that adding the comparison of the results obtained by "a stochastic process generated by computer" with at least a case results obtained from the humans might be a good idea.
  • Not all symbols are explained during the first mention (e.g., eq. (3) Ak) which makes it more difficult reading, understanding and replicability of the procedure and the results.
  • Do not artificially increase (duplicate) the number of equations, since eq. (12) is the same as eq. (6), (13) as (7), and (14) as (8). It would be enough just to refer to previously already written equations.
  • The language might be improved. In some cases, shorter sentences will make the text easier to follow (e.g., L555-559, etc.).
  • Is it necessary to have 8 self-citations?
  • The conclusion section should be improved. There are too many references mentioned there (even 2 of them mentioned for the first time in the whole manuscript).

Reviewer 4 Report

I read the paper with pleasure.

The paper compares several approaches to derive a preference vector based on pairwise comparisons in terms of statistical measures.

I have a few comments:

Line 76: "singular" --> single?

Line 144: Is T[2] redundant? if Wij can be expressed as ratios wi/wj then of course W is consistent and vice versa.

Table 1: the formula for LLSM shows the problem being solved, not a ready to us formula to compute it. For instance one may mention that the solution is obtained by first replacing log(wi) with a variable yi, solving a convex minimization problem

min 1/2 sum_i sum_j(log(aij)-yi+yj)^2

with solution y that satisfies

L  y = r, with r_i=sum_j log(aij) and L the laplacian matrix corresponding to the nonzero entries (in the general case where pairwise comparisons are incomplete, otherwise L is the Laplacian matrix of a complete graph) and then

w_llss= exp(y) where xp is the componentwise exponential

e., g. see

Bozóki, Sándor, and Vitaliy Tsyganok. "The (logarithmic) least squares optimality of the arithmetic (geometric) mean of weight vectors calculated from all spanning trees for incomplete additive (multiplicative) pairwise comparison matrices." International Journal of General Systems 48.4 (2019): 362-381.

Maybe there is a similar expression for the other methods in the table

Table 2: what are \overline{v} and \overline{w}?

What about some comparisons in terms of Kendall's correlation, to account for the ordinal differences, e.g., see

Faramondi, Luca, Gabriele Oliva, and Roberto Setola. "Multi-criteria node criticality assessment framework for critical infrastructure networks." International Journal of Critical Infrastructure Protection 28 (2020): 100338.

Reviewer 5 Report

R: reviewer (where needed)

Abstract

“There are tens of priorities deriving methods”

R: not in the abstract. Not in a scientific publication (… numerous…, … a considerable number of…)

R: judges judge but experts and/or DM assess or evaluate (also estimate)

It is known that when decision makers (DM) are consistent with their pairwise judgments about various decision options,…

R: it lacks logic (it cannot be expressed this way no matter what).

R: …. all pairwise comparisons are consistent…

This research compares a few selected PDM from the perspective of their ranking credibility which is evaluated with the application of a few 15 available statistical measures

R: This research study compares selected PDM from the perspective of their ranking credibility which is evaluated by relevant statistical measures…

R: “Mean Spearman Rank Correlation Coefficient (MSRC), Mean 16 Pearson Correlation Coefficient (MPCC), and Mean Average Absolute Deviation (MAAD).” Should be removed from Abstract.

R: “These 17 measures designate the difference between PV estimation quality; understood as a true ranking 18 preservation capability of the selected PDM.” no idea what the author wanted to express

R “Pairwise Comparison Matrices (PCM)” used w/o definition (unacceptable)

R: pairwise judgments -> pairwise comparisons

R: fundamental considerations -> theoretical considerations

Intro

Creating a ranking based on comparing alternatives in pairwise mode…

R: pairwise mode?

Creating a ranking based on pairwise comparing comparisons…

Line 45 should by follow by:

In time, numerous studies presented scientific evidence of fundamental flows In AHP. They include:

  • https://www.sciencedirect.com/science/article/abs/pii/S0377221714006171
  • https://www.researchgate.net/publication/222835380_A_critical_analysis_of_the_eigenvalue_method_used_to_derive_priorities_in_AHP
  • https://www.sciencedirect.com/science/article/abs/pii/S0377221706008538
  • https://www.semanticscholar.org/paper/Important-Facts-and-Observations-about-Pairwise-Koczkodaj-Mikhailov/a8af91fbbb83a1e302b9ba44121a5fe52eb97964
  • https://www.sciencedirect.com/science/article/abs/pii/S0022249684710340
  • https://dl.acm.org/doi/10.1515/amcs-2016-0050

T[2]: The matrix of ratios W(w)=(wi/wj) is consistent if, and only if, n is its PREV and  Ww=nw. Further, w>0 is unique to within a multiplicative constant.

R: This is taken from Saaty’s AHP publication of 1977 and W-R-O-N-G. Every matrix (w_i/w_j) for all w_i, w_j > 0 is consistent (eigenvector ha nothing to do with it).

Certainly, when the AHP is utilized, there is not an W(w) which would reflect true  weights given by the actual PV.

R: the above does not make sense. It does but only for an inconsistent PC matrix.

At the same time, it is argued that so long as inconsistency in pairwise  judgments is tolerated, the PREV is the basic theoretical concept for PCB ranking problems and no other PDM qualifies

R: the above  references provide mathematical theory and examples that it is W-R-O-N-G.

The reviewer has stopped commenting “line-by-line” but will include general comments.

The obtained results seem to be worth publishing provided the presentation is improved and Conclusions rewritten.

Round 2

Reviewer 1 Report

I reconsidered my opinion on the manuscript, but I reached the same conclusion. My conclusions are based on the data exposed in the paper which are, I maintain, inconclusive. 

Let me consider Table 6 (I consider the log-normal perturbation valid, but the other perturbations are quite arbitrary): in this table the mean absolute deviation never reaches the second decimal digit, and only in one case, it gets close to it. This means that prioritization methods, from the quality point of view, are practically identical.

Even from the ranking point of view, as shown in Table 11, they are essentially the same: the lowest correlation is 0.999989.

As the author says (l. 510-562) there have been already many studies on the subject. One might even add: Kułakowski, K., Mazurek, J., & Strada, M. (2021). On the similarity between ranking vectors in the pairwise comparison method. Journal of the Operational Research Society, 1-10.

My conclusion is that this paper presents a methedologically ok approach with nice computations, which, however, offers too small (or no) further insight into the prioritization procedures.

Author Response

I have already responded to the comment concerning the alleged inconclusiveness of the results presented in the reviewed research during my second revision of the paper. I repeat that in this version of the manuscript, discrepancies and similarities of the selected priorities deriving methods (PDMs) are examined and they are clearly presented from the statistical viewpoint. They are also valuated from the perspective of statistical theory, what yet has not been achieved in this scale by any other research. There are many tables in the paper presenting both DISCREPANCIES and SIMILARITIES of named PDMs which are statistically significant. Focusing exclusively on selected data to support one's own standpoint is pointless. Proposed Probability Distributions (PDs) of the perturbation factor ARE NOT arbitrary as the reviewer asserts. They are commonly applied for similar studies since Zahedi (1986). Yes, similar papers devoted to similar studies have been published before, and they are still published AS VALUABLE CONTRIBUTIONS TO THE SUBJECT e.g. Kułakowski, K., Mazurek, J., & Strada, M. (2021). On the similarity between ranking vectors in the pairwise comparison method. Journal of the Operational Research Society, 1-10.

Reviewer 2 Report

Dear Author,
1) With ref to my 9 Pt. , AHP is always suffering by Rank reversal Issue. To avoid that there is no provision mentioned in paper. if I removed any criteria or added criteria  then then it impacts of ranking of alternatives.  
2) In AHP Model , single DM's preferences are mentioned. But In group of DMs , how can handle it.
3) In conclusion , you should mention the limitations of this paper and how would be your method is beneficial to our society and its value added to researchers.
4) You should be compared your obtained results to the results obtained by another similar types of method .   

Author Response

Dear Reviewer:

1) With ref to my 9 Pt. , AHP is always suffering by Rank reversal Issue. To avoid that there is no provision mentioned in paper. if I removed any criteria or added criteria  then then it impacts of ranking of alternatives. 

=> Yes, I am aware of this, however, it is a different subject which I am developing concurrently, and probably I will be able to publish it very soon. For this paper, this problem is out of its scope, and do not influence obtained results, as I explained it to you in my previous rebuttal letter.

2) In AHP Model, single DM's preferences are mentioned. But In group of DMs , how can handle it.

=> This is also a different story, partially developed in: https://dx.doi.org/10.15240/tul/001/2019-4-013

However, this issue also do not influence the obtained results as there are two ways of constructing group Priority Vectors (PVs), first averaging (geometric mean) individual PVs, or averaging (geometric mean) individual pairwise comparisons for Pairwise Comparison Matrix particular entries. Because Monte Carlo Simulations applied for this research paper reflect possible errors obtained during the pairwise comparison process, they also take into account the errors obtained during group decisional processes while the second way is applied.

3) In conclusion , you should mention the limitations of this paper and how would be your method is beneficial to our society and its value added to researchers.

=> I have revised again the section "Conclusions".

4) You should be compared your obtained results to the results obtained by another similar types of method .

=> Yes, I am working on that. I am going to present the results of my other case-based examination soon. I have already collected the research material and I will examine its results in the near future. Thank you for your patience.

I can only hope that my answers did satisfy your questions and suggestions and that you will choose now to accept for publication the current version of my research paper. I thank you so much!

Reviewer 3 Report

Dear Author,

you responded to the issues raised during the previous reviewing stage. Some of them were accepted and corrected by you, while for others the explanations which might be accepted were provided. Hence, I recommend accepting this manuscript for publishing in this journal.

Author Response

Thank you very much!

Reviewer 4 Report

I am satisfied with the review

Author Response

Thank you.

Reviewer 5 Report

I see some improvements done

This manuscript is a resubmission of an earlier submission. The following is a list of the peer review reports and author responses from that submission.

Round 1

Reviewer 1 Report

I read this paper with interest as it falls in my area of interest. It is essentially a computational study on different prioritization methods.

I believe that the paper is excessively long even if I need to give credit to the author for a neat overview of the prioritization problem. Yet, I think that a shorter paper would be in the interest of the author, as too many words distract the reader from the focus of the paper. It is also difficult to grasp the relevance of a number of references, e.g. [2], [3], [4], [10], [13], [14], [15], [17], [20], [31].

Between page 3 and 4, D[3] should be presented at the beginning since the theorems are based on it. Also, consistency implies reciprocity.

I do not understand why one of the methods has the word “utility” in its name. In fact, I do not see any connection with utility or utility theory. It is also claimed that it is related to spectral theory, but I do not see that either.

The tables are extremely hard to read. What’s the meaning of boldface and the double underlining? Would not a graphical representation be better?

I fail to see a significant incremental contribution with respect to the methodology and the new findings when I compare this manuscript with previous papers by the author:

Kazibudzki, P. T. (2012). Note on some revelations in prioritization, theory of choice and decision making support methodology. African Journal of Business Management6(48), 11762-11770.

Kazibudzki, P. T. (2019). The quality of ranking during simulated pairwise judgments for examined approximation procedures. Modelling and Simulation in Engineering2019.

Also referring to the previous comment, my main problem in accepting this paper is its lack of relevance, or so it seems to me. If I asked myself if I could use the findings of this paper the next time I choose a prioritization method, at the moment, my answer is negative. Analyses of this type, as correctly pointed out by the authors, are already present in the literature and the results of this paper do not seem significant, as most of the time, they only differ at the second or third decimal digit. Said this, the paper seems quite inconclusive.
